# MESSES: Software for Transforming Messy Research Datasets into Clean Submissions to Metabolomics Workbench for Public Sharing

**DOI:** 10.3390/metabo13070842

**Published:** 2023-07-12

**Authors:** P. Travis Thompson, Hunter N. B. Moseley

**Affiliations:** 1Superfund Research Center, University of Kentucky, Lexington, KY 40536, USA; 2Department of Molecular & Cellular Biochemistry, University of Kentucky, Lexington, KY 40536, USA; 3Center for Clinical and Translational Science, Lexington, KY 40536, USA; 4Markey Cancer Center, University of Kentucky, Lexington, KY 40536, USA; 5Institute for Biomedical Informatics, University of Kentucky, Lexington, KY 40536, USA

**Keywords:** data sharing, dataset deposition, metadata capture, data transformation, Python programming language, Metabolomics Workbench

## Abstract

In recent years, the FAIR guiding principles and the broader concept of open science has grown in importance in academic research, especially as funding entities have aggressively promoted public sharing of research products. Key to public research sharing is deposition of datasets into online data repositories, but it can be a chore to transform messy unstructured data into the forms required by these repositories. To help generate Metabolomics Workbench depositions, we have developed the MESSES (Metadata from Experimental SpreadSheets Extraction System) software package, implemented in the Python 3 programming language and supported on Linux, Windows, and Mac operating systems. MESSES helps transform tabular data from multiple sources into a Metabolomics Workbench specific deposition format. The package provides three commands, extract, validate, and convert, that implement a natural data transformation workflow. Moreover, MESSES facilitates richer metadata capture than is typically attempted by manual efforts. The source code and extensive documentation is hosted on GitHub and is also available on the Python Package Index for easy installation.

## 1. Introduction

Open science is both a concept and movement to make all research data, products, and knowledge openly accessible by anyone, both promoting collaborative research efforts which can involve professionals, trainees, and non-professionals and improving the evaluation, reproducibility, and ultimately the rigor of the science [1,2]. A fundamental part of open science is the FAIR guiding principles for data management and stewardship, which focuses on making research data Findable, Accessible, Interoperable, and Reusable [3]. And the adoption of FAIR across the scientific community has spearheaded the growth of open science. Within the context of biological and biomedical research involving metabolomics and lipidomics experiments, a major goal of open science is for the resulting metabolomics and lipidomics datasets be deposited in an open data repository like Metabolomics Workbench [4] or MetaboLights [5]. Moreover, new funding agency policies are requiring deposition of research data into open scientific repositories, for example, the new National Institutes of Health (NIH) Data Management and Sharing (DMS) Policy that went into effect 25 January 2023 [6]. This new NIH DMS policy strongly promotes the deposition of “scientific data” into the most appropriate scientific repository, especially NIH-supported repositories like the Metabolomics Workbench.

While deposition of metabolomics and lipidomics datasets has become essential to satisfy both funding agency and publication requirements, there is less focus on the quality of deposition. This is partly due to the significant effort required to produce a high-quality deposition from whichever formats the data and related metadata are currently in. While metabolomics deposition reporting standards exist [7], historically, data repositories have used low deposition requirements and developed their own deposition tools with web interfaces in order to encourage deposition [4,8,9,10,11]. Moreover, these deposition tools and web interfaces were developed with the final deposition format in mind rather than the initial formats of the data and related metadata. In most cases, the data and metadata start in a tabular format within one or more spreadsheets, often with minimal organization, which must be converted into an organized format that can be handled by a given deposition system. Thus, the effort to generate a high-quality deposition is often manually intensive and quite demanding.

MESSES stands for Metadata from Experimental SpreadSheets Extraction System and originally was developed as part of a laboratory information management system (LIMS). A LIMS is essentially a database and user interface for storing, organizing, and accessing information needed to manage and document the activities in a lab. This can be inventory information, personnel information, experiment information, etc. Initially, a predecessor of MESSES was created to capture experimental data and related metadata that would go into the relational database of a LIMS. Transforming messy semi- and unstructured experiment data into a form that can be inserted into a relational database faces many of the same challenges as uploading experimental data into an online repository. Over several years, MESSES was improved, expanded, and re-implemented as a standalone package to handle this new use case.

As illustrated in Figure 1, the MESSES package enables the overall capture, validation, and conversion process using three major commands: ‘extract’, ‘validate’, and ‘convert’. The MESSES ‘extract’ command is used to transform tabular data into a representative JavaScript Object Notation (JSON) file format using a tagging system. Users provide descriptive ‘tags’ above data columns that allow MESSES to extract and interpret the data. Once extracted, the data is organized into a JSON representation. Tags can be added manually, but MESSES provides tagging automation methods to easily add tags based on header names present in the tabular data. There are also facilities to modify the data, such as changing names or removing data not needed for a particular deposition.

The ‘validate’ command evaluates whether the extracted MESSES JSON representation conforms to a specific data schema, i.e., a specific (nested) data structure with specific fieldnames and associated values with specific data types, that is needed for eventual conversion into a deposition format. The command includes sub-commands to assist with creating and validating the schema(s) used for the actual validation.

The ‘convert’ command is used to convert the MESSES JSON representation into the mwTab JSON and tab-delimited formats. MESSES can handle the heterogeneous mwTab deposition format designed for both nuclear magnetic resonance (NMR) and mass spectrometry generated (MS) datasets. Detailed documentation for installing and using MESSES is available on GitHub and package installation is straight-forward via the Python Package Index.

## 2. Materials and Methods

Figure 1 provides an overview of the data extraction, validation, and conversion workflow enabled by MESSES. This workflow starts with metadata and data in tabular format that is extracted into an intermediate MESSES JSON format which is further converted into the final mwTab deposition formats. However, the process is not expected to be error free in the beginning and MESSES provides warning and error feedback for the user at each step, especially the validation step, enabling an error correcting workflow.

### 2.1. Third Party Packages

MESSES leverages many third-party Python libraries and packages to accomplish its major tasks. MESSES uses the docopt library [12] to implement a command line interface (CLI) from a Python docstring description. Next, MESSES uses the jsonschema library to validate user JSON input against an expected schema generated by MESSES in JSON Schema format. JSON Schema is a declarative schema language for describing an expected data schema for the purpose of validating and annotating JSON representations of structured data [13,14]. JSON Schema is developed under an OpenJS Foundation [15] project with incubation status and an active growing community of users. MESSES uses the jsonschema library to perform the lion’s share of the validate command as well as to validate user input in the convert command. The submodules validate_schema.py and convert_schema.py include specific subschemas and schema templates used to generate final schemas for validation. The Protocol Dependent Schema (PD schema) and Experiment Description Specification base schema (EDS base schema) provide the bulk of the final integrated schema in JSON Schema format that is used for validation via the jsonschema library.

MESSES uses a collection of packages to work with tabular data. Specifically, pandas [16], numpy [17], and openpyxl [18] are all used to work with tabular data. The pandas package is used for reading and writing, numpy is used for optimized data access, and openpyxl and xlsxwriter are used by pandas to write Excel files. To implement matching by Levenshtein distance, the jellyfish package is used. The Cython package [19] is used to optimize and speed up some algorithms implemented with Cython language extensions that enable translation to C++ code and compilation to a compiled importable submodule. The mwtab package [8,20] is used to convert mwTab JSON format to the mwTab tab-delimited format, both developed by the Metabolomics Workbench. A list of packages and their versions are in Table 1.

### 2.2. Package Organization and Module Description

Although MESSES is primarily designed to be a command line tool, it does provide an equivalent application programming interface (API), which can be utilized if so desired. A high-level CLI that serves as an entry-point to each command is implemented in the __main__.py submodule, but each command implements its own CLI as well. Each command, extract, validate, and convert, are in their own module. The extract module contains the extract.py submodule that implements the entire extract command, with the addition of a cythonized submodule that optimizes a part of the code for the extract command. The heart of the extract module is a tag parser that identifies pound-delimited tags which direct the extraction of data from tabular files as tags and associated data are parsed.

The validate module contains the validate.py submodule that implements the validate command and the validate_schema.py submodule that simply holds the built-in schemas and schema templates in JSON Schema format for the command. The convert module is broken into more pieces. The convert.py submodule implements the convert command, the convert_schema.py submodule holds the schemas and schema templates in JSON Schema format for the command, the user_input_checking.py submodule validates conversion directives, and there are submodules for the built-in conversion directives and specific code for each supported conversion format. Table 2 lists the submodules of MESSES, Figure 2 shows a module diagram, and Figure A1 shows a directory tree of the source code.

### 2.3. Tagging System

In order to extract organized data from arbitrarily placed and organized data tables within a spreadsheet in a programmatic way, some kind of system has to be devised. This could be something as simple as requiring a given data table be on the very first sheet row and for the starting row to have column names for every column or columns in a certain order; however, this type of implementation would be very fragile. Therefore, we decided to create a more robust system that could handle more complicated and/or arbitrary data arrangements and reduce the verbosity to a minimum. The system we devised uses an extra layer of tags inserted into an existing data spreadsheet at specific locations that tell the extract command how to transform the data sections of the sheet (i.e., data tables) into key-based records representable in both JSON format and a relational database.

This initial system served its function well, but it became clear that more functionality was sorely needed: (i) a way to programmatically add tags to sections of tabular data within a sheet and (ii) a way to modify field values. So, the system was expanded to provide facilities to do both. Ultimately, there are three parts to the tagging system that are distinct from one another but have similar syntax and ideas. The “export” part involves “export” tags that are directly inserted into an existing sheet before a section of tabular data. It is the base system that must be used for the extraction to work at all. The “automation” part is used to automate adding “export” tags to tabular data. Based on the header values in your data, you can use “automation” tags to insert (add) the “export” tags automatically. A good use case for automation is when you have data generated by a program in a consistent way. Instead of manually adding export tags to the program output each time, you can create an “automation” spreadsheet that will add the “export” tags for you. The last “modification” part is used to modify record values. It can be used to prepend, append, delete, overwrite, or regex substitute values. An example use-case would be to update old naming conventions. Validly tagged files in their tabular or JSON form can be referred to as directives as they direct the extraction (automate, export, and modify) actions of MESSES. To reduce confusion between tags and directives, “tags” generally refer to the extra text added above a specific table, while “directives” are the tags and the associated table taken as a whole. Each row of a tagged table is an individual directive.

Each part of the tagging system must be in their own sheet or file for the extract command. By default, export tags are expected in a sheet named ‘#export’, if given an Excel file without specifying a sheet name. If given a CSV file, then this file is expected to have export tags. Modification tags are expected in a sheet named ‘#modify’ by default but can be specified using the --modify option. The option is very flexible and can be used to specify either a different sheet name in the given Excel file, a different Excel file, a different Excel file with a different sheet name, a Google Sheets file, a Google Sheets file with a different sheet name, a JSON file, or a CSV file. Automation tags are similarly specified using the --automate option or otherwise expected in a sheet named ‘#automate’ by default. More detailed descriptions and examples of the tagging system can be found in the package documentation.

### 2.4. MESSES JSONized Data and Metadata Representation

The data schema developed for MESSES was designed to capture generalized experimental descriptions and data in an abstract way. To handle the arbitrary number of fields that widely varying experimental datasets would have, the schema supports multiple integrated entity–attribute–value (EAV) models. It is organized into several tables with a unique record identifier and a flexible collection of fields, with certain fields having a descriptive attribute relationship with another field. Note that we use the term “table” to refer to the JSON object of the same name. A “record” would be a named element inside a “table”, which would normally correspond to a row in a spreadsheet table. A “field” would be a named element inside a “record”, which would normally correspond to a column in a spreadsheet table.

There are 6 tables: project, study, protocol, entity, measurement, and factor.

A project generally refers to a research project with multiple analytical datasets derived from one or more experimental designs.
○The project table entries would have information about the project, such as PI contact information and a description of the project.A study is generally one experimental design or analytical experiment inside of the project.○The study table entries would have information about each study, such as PI contact information and a description of the study.A protocol describes an operation or set of operations done on a subject or sample entity.○The protocol table entries would have information about each protocol, such as a description of the procedure and details about the equipment used.Entities are either subjects or samples that were collected or experimented on. ○The entity table entries would have information about each entity, such as sex and age of a subject or weight and units of weight of a sample. These latter examples demonstrate a descriptive attribute relationship between the weight field and the units of weight field typically indicated by ‘weight%unit’ used as the field name for units of weight.A measurement is typically the results acquired after putting a sample through an assay or analytical instrument such as a mass spectrometer or nuclear magnetic resonance spectrometer as well as any data calculation steps applied to raw measurements to generate usable processed results for downstream analysis. ○The measurement table entries would have information about each measurement, such as intensity, peak area, or compound assignment.A factor is a controlled independent variable of the experimental design. Experimental factors are conditions set in the experiment. Other factors may be other classifications such as male or female gender.○The factor table entries would have information about each factor, such as the name of the factor and the allowed values of the factor.

Figure 3 shows a lean example MESSES JSON with all of the tables, and Table 3 summarizes the descriptions and entry information for table entries.

There are additional constraints within the tables. Protocols must be one of five types: treatment, collection, sample_prep, measurement, or storage.

A treatment protocol describes the experimental factors performed on subject entities.○For example, if a cell line is given 2 different media solutions to observe the different growth behavior between the 2, then this would be a treatment type protocol.A collection protocol describes how samples are collected from subject entities.○For example, if media is taken out of a cell culture at various time points, this would be a collection protocol.A sample_prep protocol describes operations performed on sample entities.○For example, once the cells in a culture are collected, they may be spun in a centrifuge or have solvents added to separate out protein, lipids, etc.A measurement protocol describes operations performed on samples to measure features about them.○For example, if a sample is put through a mass spectrometer or into an NMR.A storage protocol describes where and/or how things (mainly samples) are stored.○This was created mostly to help keep track of where samples were physically stored in freezers or where measurement data files were located on a share drive.

Another constraint involves how subjects and samples inherit or derive from each other.

If a sample comes from a sample, it must have a sample_prep type protocol.If a sample comes from a subject, it must have a collection type protocol.Subjects should have a treatment type protocol associated with it.

### 2.5. Testing

The MESSES package was originally developed in a Linux operating system (OS) environment but has been directly tested on Linux, Windows, and MacOS operating systems. Each module and submodule include unit-tests that test all critical functions. Every function in every module is tested to make sure it gives the expected output when it should and errors when it should. Every command and associated command line option are tested, for example, the update and override options for the convert command. Testing is automated using GitHub Actions. Total testing code coverage for the MESSES package is above 90%.

## 3. Results

### 3.1. The Command Line Interface and Overall Metabolomics Workbench Deposition Workflow

The MESSES CLI has a delegated implementation. In other words, there are four separate CLIs, one for each command and one main CLI. The main CLI serves as a gateway to the three commands that perform the bulk of the work and have their own CLIs. Once installed, a call to “messes --help” in the system terminal will show the gateway CLI, and calls to “messes [command] --help” will show the CLI for the selected command. Figure 4 and Figure A2, Figure A3 and Figure A4 show the main CLI, the extract CLI, the validate CLI, and the convert CLI, respectively.

The MESSES CLI was designed with a great deal of flexibility, anticipating users’ desire to use the software in unpredictable ways. However, Figure 1 illustrates the overall workflow, using the three main commands with the intention of creating a deposition to Metabolomics Workbench. Starting from the assumption that all data files are untagged, the first step would be to add tags to the data so it will be exported into the MESSES JSON format correctly. Tags can be added manually or with automation directives used by the extract command (i.e, tagging step). Modification directives can also be used to modify the data as necessary for tasks such as renaming. Once tagged, the extract command extracts and exports the (meta)data into a MESSES JSON file. You may have to fix some errors if you have malformed tags or directives. Next, take the exported MESSES JSON file and deliver it to the validate command. It is recommended to use the --format option and specify “mwtab”. It is also recommended to create a protocol-dependent schema and use the --pds option with the schema to perform additional validation. A protocol-dependent schema is provided in the Appendix A. There will likely be warnings and errors after running the validate command, and they should be corrected in the data. After correcting the errors and warnings, re-export the MESSES JSON with the extract command and re-validate with the validate command until there are no more errors or warnings of concern. Once the MESSES JSON file validates with no errors or warnings, deliver it to the convert command. Use the mwtab sub-command and select the appropriate machine type for your data, ms, nmr, or nmr_binned. The convert command should output a mwTab JSON and tab-delimited file. But even with a clean validation, it is still possible to have some errors that prevent conversion. If there are errors, correct them and start from the extraction step again.

### 3.2. Creation of an Example Mass Spectrometry Deposition

We demonstrate the capabilities of MESSES with a paired down example based on an ion chromatography Fourier transform mass spectrometry (IC-FTMS)-targeted metabolomics dataset of mouse colon tissue already deposited into Metabolomics Workbench Study ST001447 [21] using an earlier prototype of MESSES. Although this dataset was previously uploaded using an earlier version of MESSES, what is demonstrated here is using the latest version. This demonstration walks through the (meta)data extraction from Excel spreadsheets, JSON validation, and conversion steps to produce a deposition-compliant dataset in both the mwTab JSON and tab-delimited formats. Note that the figures below are general truncated examples. There are full examples with package commands and description that transform real datasets, available in the Appendix A and in the examples directory of the GitHub repository.

#### 3.2.1. Extraction from Spreadsheets

Figure 5 shows screenshots of the executed command and directory of files when running the extract command. The metadata Excel spreadsheet has metadata for several tissues besides colon, which are removed with the ‘--delete’ option. Likewise, certain unrelated protocols (acetone_extraction and lipid_extraction) involving other related analytical measurements are likewise removed. Figure 6 and Figure 7 show screenshots of the metadata and measurement data Excel files used with the extract command, respectively. Note that the “#export” sheet is what the command will use by default. Figure 6 shows the original sheet with its formatting and tags added, but the “#export” sheet is a copy that removes formatting. Figure 8 and Figure 9 show screenshots of the automation and modification tags for the measurement data in separate ‘#automate’ and ‘#modify’ sheets, respectively. The automation tags are used to add export tags internally and the “#export” spreadsheet created can be saved out using the --save-export option. The modification tags are used to modify the data after it has been extracted from the spreadsheet to a JSONized form. Figure 10 shows portions of the extracted JSON organized in separate JSON objects which are represented as dictionaries in Python. The ‘entity’ dictionary describes individual subjects (mice in this instance) and individual samples derived from the subjects. The ‘factor’ dictionary describes the experimental design in terms of individual experimental factors. The ‘protocol’ dictionary describes individual protocols used in the experiment. The ‘measurement’ dictionary describes individual peak measurements derived from an IC-FTMS spectrum collected per sample. The ‘project’ and ‘study’ dictionaries describe the research project and specific study performed, including the contact and institution that the deposition comes from.

#### 3.2.2. Validation of Extracted Data and Metadata

After extraction, a user should use the validate command on the (JSON) output to validate the result. Typically, both the extract and validate commands will be used iteratively with dataset revision until no more errors or warnings are detected during validation, creating a combined extraction and validation process. If extraction involves datasets generated in a consistent format from other programs, this could essentially become an automated process; however, given the nature of most analytical labs and core facilities, a semi-automated process is expected in most cases. But by following Good Laboratory Practice (GLP) on Data Integrity [22], this semi-automated process should approach a fully automated process, especially if tagged spreadsheet templates are used for manual data collection steps. Figure 11 shows screenshots of the executed command and directory of files when running the validate command. The json subcommand identifies the extracted_result.json as being in JSON format. The ‘--pds’ option identifies the specific (protocol-dependent) PD schema to validate against. The ‘--format mwtab’ option indicates the conversion format specific schema to validate against. The ‘--silent nuisance’ option ignores common warnings that most often can be ignored. Figure 12 shows a portion of this PD schema used here, and Figure A5 shows a portion of this PD schema transformed into JSON Schema. This example is clean and complete and thus does not show any warnings or errors during validation. However, Figure A6 and Figure A7 demonstrate common warnings and errors that often occur.

#### 3.2.3. Conversion into mwTab Formats

Once the extracted MESSES JSON is validated, it can be converted into the mwTab JSON and tab-delimited formats. Figure 13 shows screenshots of the executed command and directory of files when running the convert command. The ‘mwtab ms’ subcommand identifies the output type, which is followed by the input extracted_results.json filename and the output filename without file extension. Two separate output files are generated in mwTab JSON (output.json) and tab-delimited (output.txt) formats. Technically, the mwTab JSON format is generated first and then the mwtab library is used to convert it further into the mwTab tab-delimited format. Figure 14 and Figure 15 show screenshots of portions of the mwTab JSON and tab-delimited text outputs, respectively. Note that the ANALYSIS_ID and STUDY_ID default to 000000. Before submission to the Metabolomics Workbench these need to be updated manually with the IDs they give you, or they can be updated by using the --update option to update that portion of the conversion directives.

## 4. Discussion

MESSES is a useful tool for turning messy, disorganized data and metadata into the proper format for deposition into Metabolomics Workbench. MESSES and its prior prototypes have been used to deposit over 40 studies into Metabolomics Workbench (see Table A1), many of which provide the richest level of metadata demonstrated so far in dataset deposition into Metabolomics Workbench. MESSES was designed to improve deposition quality and metadata consistency, which are known issues in scientific repositories like Metabolomics Workbench [8,9]. The package provides a way to organize, filter, and modify data so that it can be put into the proper form, and its automation support makes adding MESSES into workflows much easier. Although a significant amount of time and effort went into refining the package so that it is as easy to use and understand as possible, there is some intellectual overhead required to initially setup all the tags, validation schemas, and conversion directives. Additional supportive sub-commands are included where applicable to make learning and troubleshooting the tool easier for new users. Also, there is extensive documentation available to help with the learning curve: https://moseleybioinformaticslab.github.io/MESSES/ (accessed on 30 June 2023). In addition, when installed via the Python package management system pip, a console script “messes” is created automatically for the user, providing easy access to the CLI.

The package has been developed in a way such that additional formats can be added into the list of inherently supported formats. But the package is also generalized enough that anyone should be able to use it to convert to whatever arbitrary format is desired, as long as it has a JSON representation. Going from the JSON representation to another non-JSON representation would have to be done using another tool if the format is not supported in MESSES. Currently, only the mwTab format is directly supported, but as the tool is used to create more diverse depositions, it is likely that more formats will be added. Another notable limitation is that deeply nested JSON structures cannot be created using MESSES without supplying your own Python code for the convert command. This is due to a desire to keep tags and directives simple enough to be in a tabular form, but if there is enough demand or need for deeper nesting, the tags and directives can be expanded.

## 5. Conclusions

The MESSES Python package enables a straight-forward mwTab deposition creation process that involves iterative extraction-validation steps followed by a final conversion step. MESSES was developed to help solve the specific deposition problems we faced in helping collaborators deposit their data, and we believe it can help many others with their depositions. While there is an initial learning curve, once a user sets up the needed tagging directives and validation schemas, repetitive generation of mwTab formatted depositions should be much easier. Moreover, MESSES enables a more comprehensive extraction of metadata to promote FAIRer depositions into Metabolomics Workbench.

## Figures and Tables

**Figure 1 metabolites-13-00842-f001:**
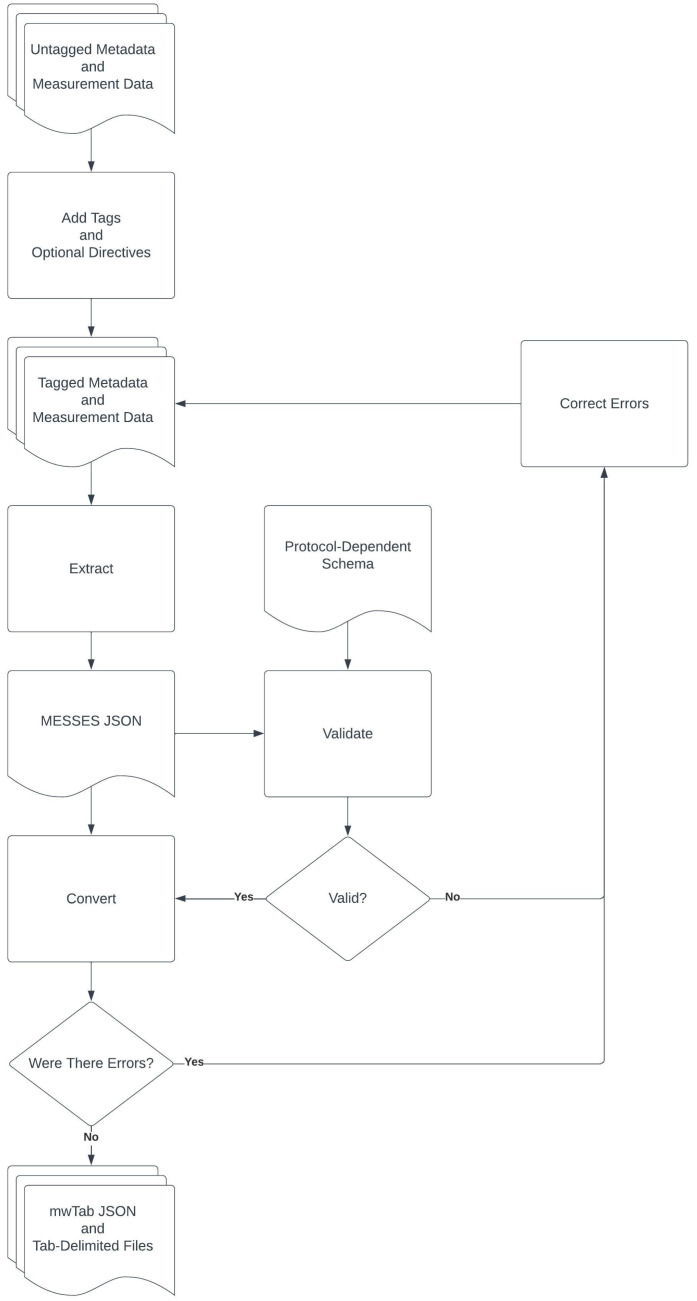
MESSES Overall Workflow Diagram. This includes each of the major steps: Extract, Validate, and Convert, along with error and warning correction steps represented by Correct Errors.

**Figure 2 metabolites-13-00842-f002:**
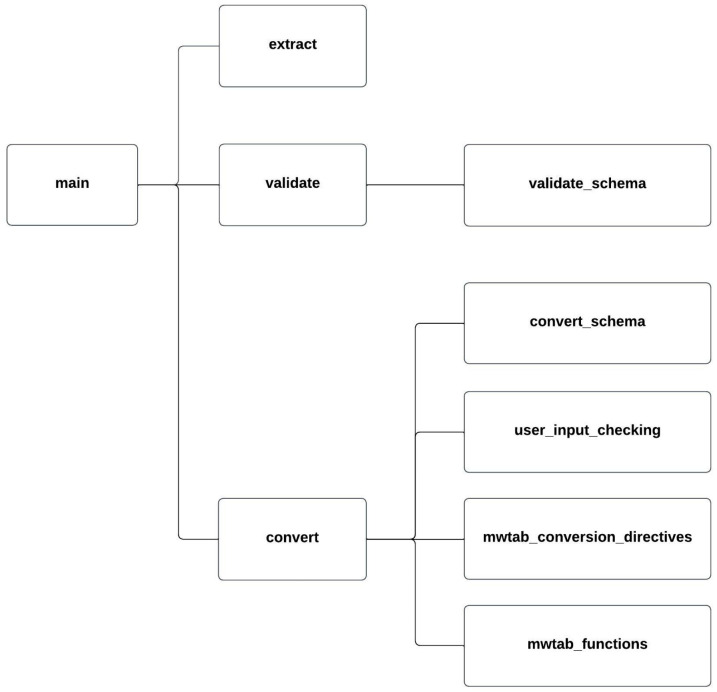
MESSES Module Diagram. Submodule and module dependencies are illustrated by connecting lines.

**Figure 3 metabolites-13-00842-f003:**
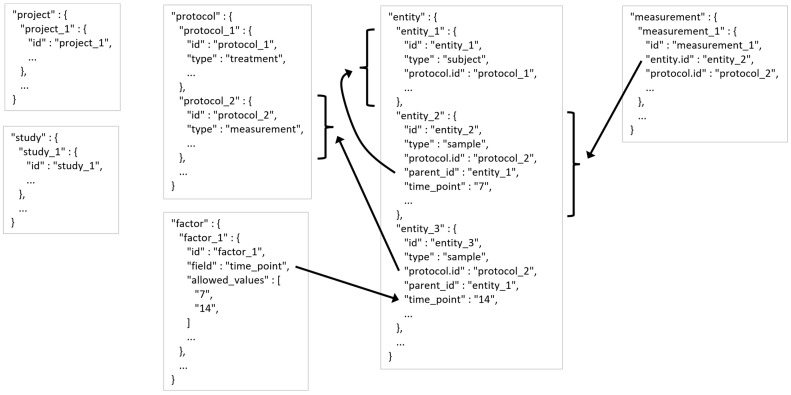
MESSES JSON Tables Example. Ellipses indicate that there could be more fields or records, while arrows point to records or fields that a field is referencing.

**Figure 4 metabolites-13-00842-f004:**
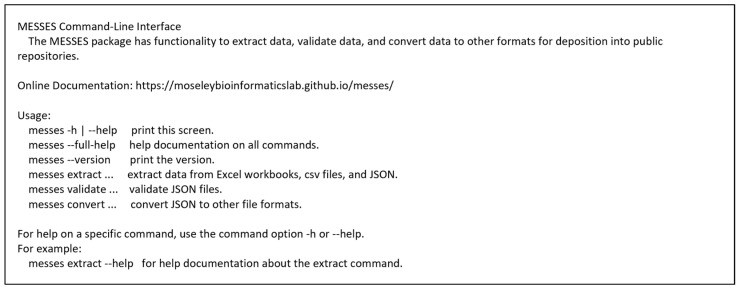
MESSES main Command Line Interface (CLI). The extract, validate, and convert commands represent distinct steps in the overall MESSES workflow.

**Figure 5 metabolites-13-00842-f005:**
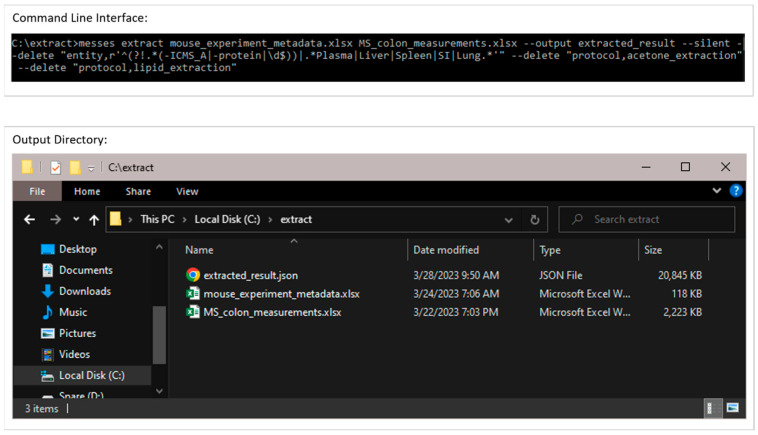
Example execution of the extract command in a Windows Command Prompt. The resulting output directory is shown in the Windows folder at the bottom.

**Figure 6 metabolites-13-00842-f006:**
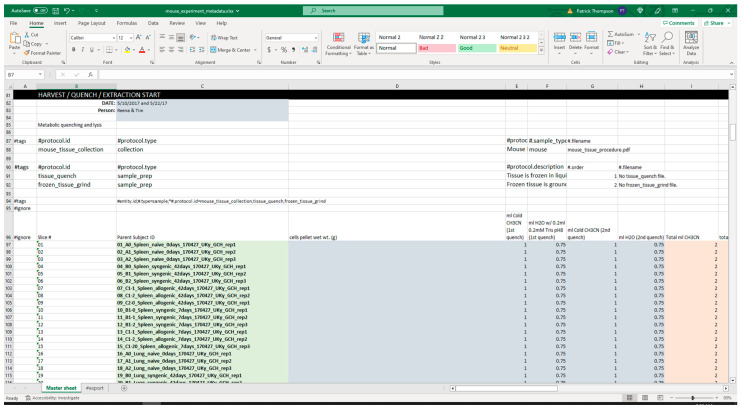
Screenshot of a portion of the metadata spreadsheet used with the extract command.

**Figure 7 metabolites-13-00842-f007:**
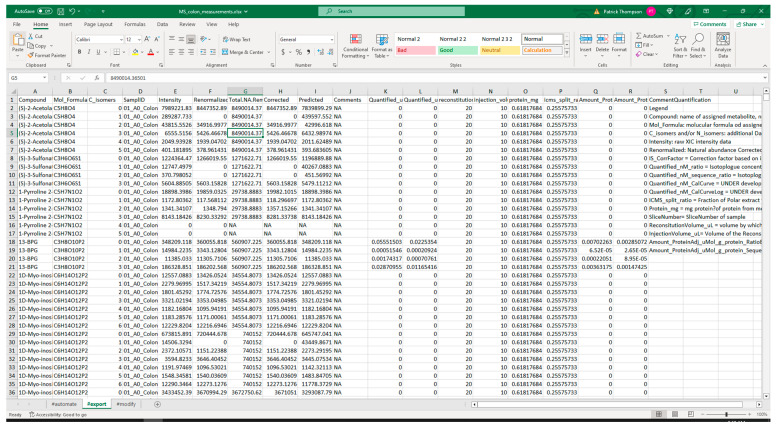
Screenshot of a portion of the measurement spreadsheet data used with the extract command.

**Figure 8 metabolites-13-00842-f008:**
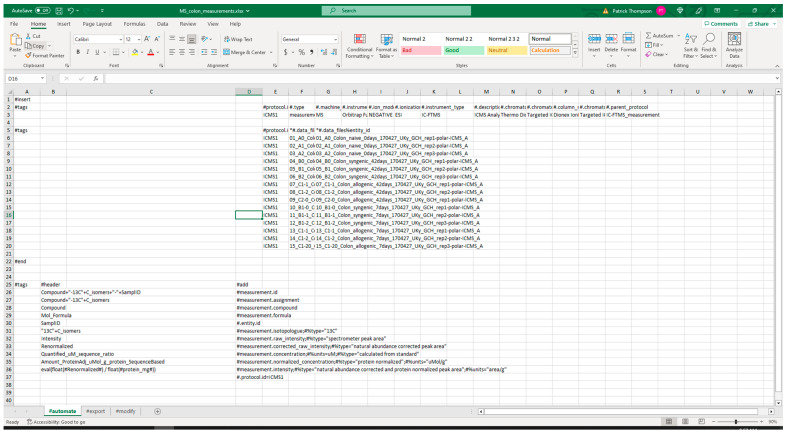
Screenshot of the automation tags used with the measurement data when executing the extract command.

**Figure 9 metabolites-13-00842-f009:**
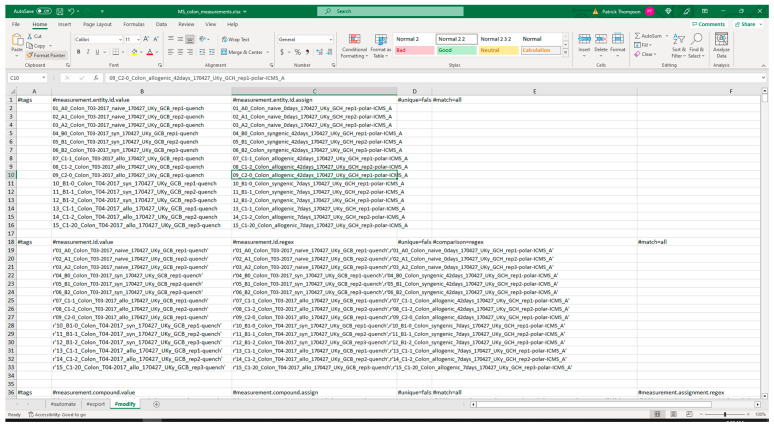
Screenshot of a portion of the modification tags used with the measurement data when executing the extract command.

**Figure 10 metabolites-13-00842-f010:**
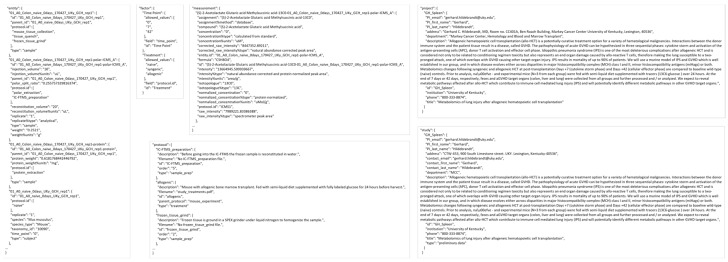
Portions of each table in the MESSES JSON file output generated by the extract command.

**Figure 11 metabolites-13-00842-f011:**
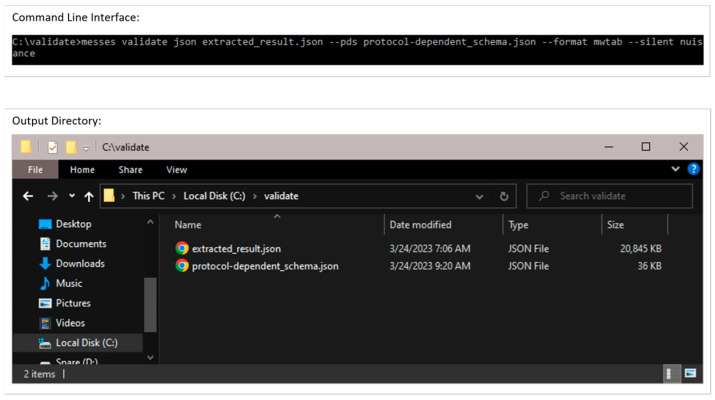
Example execution of the validate command in a Windows Command Prompt. The resulting output directory is shown in the Windows folder at the bottom.

**Figure 12 metabolites-13-00842-f012:**
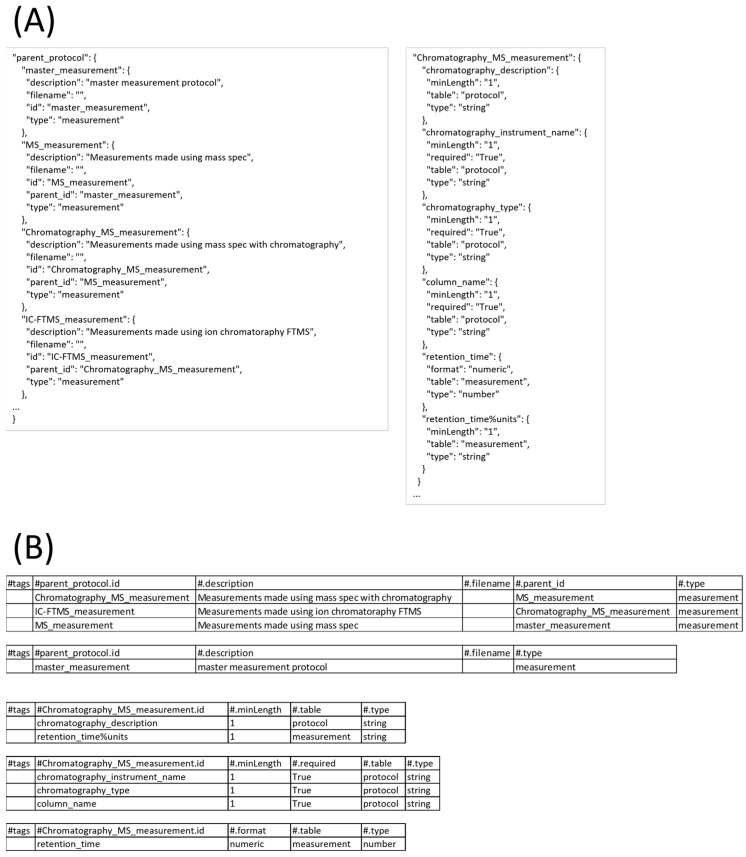
A portion of the parent_protocol table and protocols in the protocol-dependent schema. (**A**) is in JSON format and (**B**) is in tagged tabular format.

**Figure 13 metabolites-13-00842-f013:**
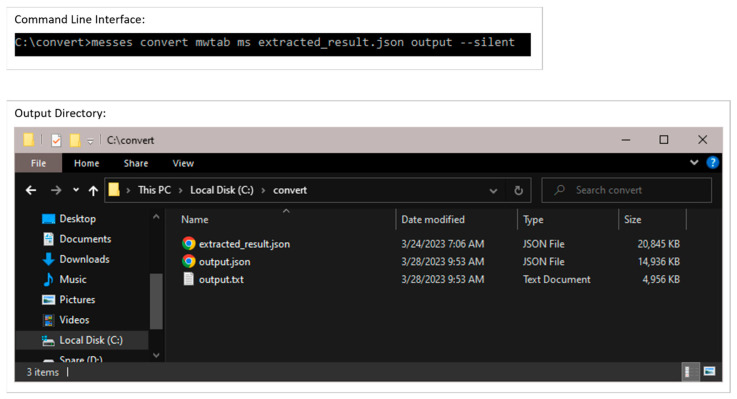
Example execution of the convert command in a Windows Command Prompt. The resulting output directory is shown in the Windows folder at the bottom.

**Figure 14 metabolites-13-00842-f014:**
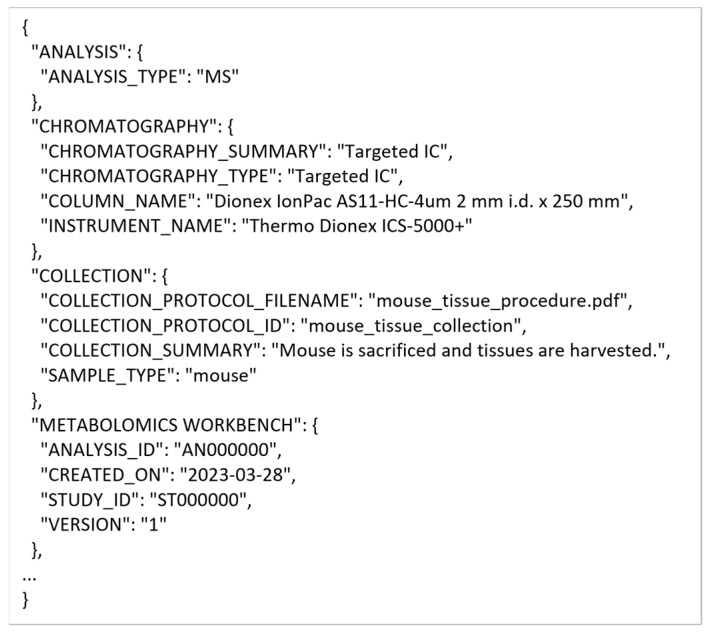
A portion of the mwTab JSON output generated by the convert command.

**Figure 15 metabolites-13-00842-f015:**
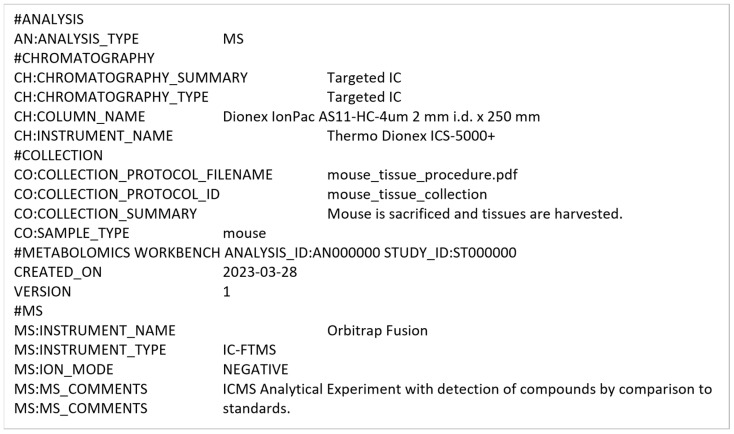
A portion of the mwTab tab-delimited text output generated by the convert command.

**Table 1 metabolites-13-00842-t001:** Library dependencies for MESSES.

Package	Version	Utilization	PyPI URL ^a^
docopt	0.6.2	Implement CLI.	https://pypi.org/project/docopt/
jsonschema	3.0.1	Validate JSON files.	https://pypi.org/project/jsonschema/
pandas	0.24.2	Read and write tabular files.	https://pypi.org/project/pandas/
numpy	1.22.4	Optimize tabular data algorithms.	https://pypi.org/project/numpy/
openpyxl	2.6.2	Write Excel files.	https://pypi.org/project/openpyxl/
xlsxwriter	3.0.3	Write Excel files.	https://pypi.org/project/xlsxwriter/
jellyfish	0.9.0	Calculate Levenshtein distance.	https://pypi.org/project/jellyfish/
Cython	3.0.0a11	Optimize algorithms.	https://pypi.org/project/Cython/
mwtab	1.2.5	Create mwTab formatted files.	https://pypi.org/project/mwtab/

^a^ Accessed on 1 January 2023.

**Table 2 metabolites-13-00842-t002:** Submodules of MESSES.

Submodule	Description
__main__.py	Contains the top-most CLI.
extract.py	Implements the extract command and CLI.
cythonized_tagSheet.pyx	Cythonized version of the tagSheet method for extract.
validate.py	Implements the validate command and CLI.
validate_schema.py	Contains the JSON Schema schemas used by the validate command.
convert.py	Implements the convert command and CLI.
convert_schema.py	Contains the JSON Schema schemas used by the convert command.
user_input_checking.py	Validates conversion directives for the convert command.
mwtab_conversion_directives.py	Contains the built-in conversion directives for the mwTab format.
mwtab_functions.py	Contains functions specific to creating the mwTab format.

**Table 3 metabolites-13-00842-t003:** MESSES JSON Table Entry Summary. The “Entry Information” column is not exhaustive and simply presents examples of what kinds of information could be associated with each entry.

Table	Entry Description	Entry Information
project	A research project with multiple analytical datasets derived from one or more experimental designs.	PI Name PI Contact Information Institution Name Address Department Description Title
study	One experimental design or analytical experiment inside of the project.	PI Name PI Contact Information Institution Name Address Department Description Title
protocol	An operation or set of operations done on a subject or sample entity.	Description Type Instrument Settings Instrument Information Software Settings Software Information Data Files Generated File Detailing Protocol
entity	Either subjects or samples that were collected or experimented on.	Type Weight Sex Protocols Underwent Parent Entity Experimental Factor
measurement	The results acquired after putting a sample through an assay or analytical instrument such as a mass spectrometer or nuclear magnetic resonance spectrometer as well as any data calculation steps applied to raw measurements to generate usable processed results for downstream analysis.	Measurement Protocol Measurements Acquired Associated Entity Calculations or Statistics Labels Obtained from Measurements
factor	A controlled independent variable of the experimental design. Conditions set in the experiment. May be other classifications such as male or female gender.	Discrete Values of the Factor Units of the Values Name of Factor Field Name of Factor

## Data Availability

The Python package is available on the Python Package Index (https://pypi.org/project/messes accessed on 30 June 2023) and on GitHub (https://github.com/MoseleyBioinformaticsLab/MESSES accessed on 30 June 2023), along with extensive end-user documentation (https://moseleybioinformaticslab.github.io/MESSES/ accessed on 30 June 2023). Examples of dataset capture, validation, and conversion into mwTab deposition format are available on Figshare: https://figshare.com/articles/dataset/MESSES_Supplemental_Material/23148224 (accessed on 30 June 2023).

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
