# Peer review of "MESSES: Software for Transforming Messy Research Datasets into Clean Submissions to Metabolomics Workbench for Public Sharing"

_metabolites, 2023, doi:10.3390/metabo13070842_

Round 1

Reviewer 1 Report

The paper presents the software MESSES for transforming messy research datasets into clean submissions to metabolomics workbench for public sharing. Despite the huge amount of information, the paper is well-organized and logically presented. However, the paper needs some additional efforts before to be considered for publishing in such an esteemed journal as Metabolites.

1.      Starting from the end to the beginning, I would suggest a list of abbreviations to be provided.

2.      I would suggest the explanation of the abbreviation MESSES to appear in the abstract as well.

3.      In Section 3.2, rows 332-335, “We demonstrate the capabilities of MESSES with a paired down example …. using an earlier prototype of MESSES.” is a bit confusing... The first question that arose is why the authors have decided to present results from “an earlier prototype of MESSES”, why not with the current version?

4.      Please pay special attention the figures titles to come with the figures, especially after Figure 9. Some titles appear before the figures themselves, while the other – after them. In addition, Figure A3 appears in the text before Figure A1 and A2 – consider possible renumbering

5.      Please remove the redundant partс from the template.

6.      Refs. 12 and 21 should be completed with the access date. The access date of Refs. 14 and 18 is 2014 and 2016, respectively – need to be checked and updated.

Author Response

Reviewer 1:

The paper presents the software MESSES for transforming messy research datasets into clean submissions to metabolomics workbench for public sharing. Despite the huge amount of information, the paper is well-organized and logically presented. However, the paper needs some additional efforts before to be considered for publishing in such an esteemed journal as Metabolites.

Response:

We thank the reviewer for their positive comments.  MESSES represents many years of development, redesign, and reimplementation.

Issue 1:

Starting from the end to the beginning, I would suggest a list of abbreviations to be provided.

Response:

A list of abbreviations has been added at the beginning of the Introduction as a subsection. However, this subsection may not conform to the journal’s format.  The “Instructions to Authors” indicate that abbreviations “should be defined in parentheses the first time they appear in the abstract, main text, and in figure or table captions.”  So, the journal may opt to remove this subsection of the Introduction.

Issue 2:

I would suggest the explanation of the abbreviation MESSES to appear in the abstract as well.

Response:

            The MESSES acronym has been added after its first appearance in the abstract.

Issue 3:

In Section 3.2, rows 332-335, “We demonstrate the capabilities of MESSES with a paired down example …. using an earlier prototype of MESSES.” is a bit confusing... The first question that arose is why the authors have decided to present results from “an earlier prototype of MESSES”, why not with the current version?

Response:

We can understand the confusion with this statement. These datasets were originally deposited into Metabolomics Workbench using a MESSES prototype, but the examples presented in the paper and supplementary materials were all created with the latest version. Although what is generated by the latest version is slightly different from what was initially uploaded, the original upload is correct and accurate, so there is no need to update what has already been deposited and vetted by Metabolomics Workbench. We have added a clarifying sentence to indicate that the examples were created with the latest version:

“We demonstrate the capabilities of MESSES with a paired down example based on an ion chromatography Fourier transform mass spectrometry (IC-FTMS) targeted metabolomics dataset of mouse colon tissue already deposited into Metabolomics Workbench Study ST001447 [21] using an earlier prototype of MESSES. Although this dataset was previously uploaded using an earlier version of MESSES, what is demon-strated here is using the latest version.”

Issue 4:

Please pay special attention the figures titles to come with the figures, especially after Figure 9. Some titles appear before the figures themselves, while the other – after them. In addition, Figure A3 appears in the text before Figure A1 and A2 – consider possible renumbering…

Response:

There appears to be an issue on Metabolites end after submission, as the version we uploaded does not have this issue with the figures. We have corrected it, but the issue may reoccur when we resubmit. However, we are confident it will not make it past the proofing step. The appendix figures have been reordered to match their first mention in the text.

Issue 5:

Please remove the redundant partс from the template.

Response:

            We are unclear what the reviewer is talking about. What is part c? What is the template?

Issue 6:

Refs. 12 and 21 should be completed with the access date. The access date of Refs. 14 and 18 is 2014 and 2016, respectively – need to be checked and updated.

Response:

Fixed.

Reviewer 2 Report

This manuscript shows a new method to normalise metabolomics data and increase their usefulness in a future-present context of multiomics science.

I only have one question about the foreseeable variability of the data relate to the platforms. Did the authors think about some standard normalization of the this data? 

Author Response

Reviewer 2:

This manuscript shows a new method to normalize metabolomics data and increase their usefulness in a future-present context of multiomics science.

I only have one question about the foreseeable variability of the data relate to the platforms. Did the authors think about some standard normalization of the this data?

Response:

Normalization comes in several forms. We assume the reviewer is referring to format consistency and not data normalization, which is beyond the scope of this manuscript. We identified mwTab consistency issues in the following prior publication and preprint, of which the latter is currently under review.

Christian D. Powell and Hunter N.B. Moseley. "The mwtab Python library for RESTful Access and Enhanced Quality Control, Deposition, and Curation of the Metabolomics Workbench Data Repository" Metabolites 11, 163 (2021).

Christian D. Powell and Hunter N.B. Moseley. "The Metabolomics Workbench File Status Website: A Metadata Repository Promoting FAIR Principles of Metabolomics Data" bioRxiv 2022.03.04.483070 (2022).

However, the reviewer’s point is well taken, and we have added the following in the discussion to highlight how MESSES can improve format consistency and thus deposition quality:

“MESSES was designed to improve deposition quality and metadata consistency, which are known issues in scientific repositories like Metabolomics Workbench [8,9].” 

Reviewer 3 Report

The manuscript sounds very interesting and has merit in publication. However, there are too many figures provided in the manuscript resulting in clutter. I recommend authors revise  the number of figures/illustrations into subplots or consider important ones and provide detailed captions.

The English language presented in the manuscript is readable. However,  it can be improvized.

Author Response

Reviewer 3:

The manuscript sounds very interesting and has merit in publication. However, there are too many figures provided in the manuscript resulting in clutter. I recommend authors revise  the number of figures/illustrations into subplots or consider important ones and provide detailed captions.

Response:

Several of the figures already have multiple sections.  And this is an open-access journal with no page limitations, so there is no issue with length. So respectfully, we think the current set of figures are descriptive, even if a little over descriptive. Also, the figures will flow better with the text once they are professionally laid out in the proofs.

But as per the reviewer’s point, we have expanded several of the figure legends to be more descriptive.

Round 2

Reviewer 1 Report

The revised version of the paper addressed almost all of my points of concern.

There are just two things to be considered:

1.      The authors have supplied a list of abbreviations at the beginning of the Introduction. As they also mentioned, it might be removed by the Editors from the Introduction. So I would suggest the list of abbreviation to appear at the beginning of the Appendices.

2.      Concerning my remark of “Please remove the redundant parts from the template” – I have in mind rows 532-534, e.g.

Author Response

Reviewer 1:

The revised version of the paper addressed almost all of my points of concern.

Response:

We tried to address it all in the first round.  ;)

Issue 1:

There are just two things to be considered:

  1. The authors have supplied a list of abbreviations at the beginning of the Introduction. As they also mentioned, it might be removed by the Editors from the Introduction. So I would suggest the list of abbreviation to appear at the beginning of the Appendices.

Response:

We think it is better at the beginning of the introduction, which is what the reviewer originally suggested.  Let’s first find out if the journal will let it be in the introduction.  If the journal requires it to be moved or removed, then we will add it into the Appendices.

Issue 2:

  1. Concerning my remark of “Please remove the redundant parts from the template” – I have in mind rows 532-534, e.g.

Response:

With the version of the manuscript we have downloaded from the journal, lines 532-534 involve the “Informed Consent Statement” and the “Data Availability Statement”.  Technically, we see no redundant template to remove in these lines.  However, after reading the finer details of the Author Instructions, we see that the “Institutional Review Board Statement” and the “Informed Consent Statement” are optional if the study does not involve humans and animals: “You might also choose to exclude this statement if the study did not involve humans or animals.”  So we have removed both statements.  Also, the “Acknowledgements” was template, which we now have used to acknowledge the effects of the MW team to maintain and expand the repository:

“The authors would like to acknowledge the large continual effort that Shankar Subramaniam, Eoin Fahy, and the whole MW/UC San Diego team have put into maintaining and expanding the repository.”